# High-harmonic generation in Weyl semimetal β-WP$_2$ crystals

Yang-Yang Lv [1,5], Jinlong Xu [1,5✉], Shuang Han[1], Chi Zhang[1], Yadong Han[2], Jian Zhou [1], Shu-Hua Yao [1✉], Xiao-Ping Liu[3], Ming-Hui Lu [1], Hongming Weng [4], Zhenda Xie [1✉], Y. B. Chen [1✉], Jianbo Hu[2], Yan-Feng Chen [1] & Shining Zhu [1]

As a quantum material, Weyl semimetal has a series of electronic-band-structure features, including Weyl points with left and right chirality and corresponding Berry curvature, which have been observed in experiments. These band-structure features also lead to some unique nonlinear properties, especially high-order harmonic generation (HHG) due to the dynamic process of electrons under strong laser excitation, which has remained unexplored previously. Herein, we obtain effective HHG in type-II Weyl semimetal β-WP$_2$ crystals, where both odd and even orders are observed, with spectra extending into the vacuum ultraviolet region (190 nm, 10th order), even under fairly low femtosecond laser intensity. In-depth studies have interpreted that odd-order harmonics come from the Bloch electron oscillation, while even orders are attributed to Bloch oscillations under the "spike-like" Berry curvature at Weyl points. With crystallographic orientation-dependent HHG spectra, we further quantitatively retrieved the electronic band structure and Berry curvature of β-WP$_2$. These findings may open the door for exploiting metallic/semimetallic states as solid platforms for deep ultra-violet radiation and offer an all-optical and pragmatic solution to characterize the complicated multiband electronic structure and Berry curvature of quantum topological materials.

[1] National Laboratory of Solid State Microstructures, School of Physics, School of Electronic Science and Engineering, College of Engineering and Applied Sciences, and Collaborative Innovation Center of Advanced Microstructures, Nanjing University, 210093 Nanjing, China. [2] Laboratory for Shock Wave and Detonation Physics, Institute of Fluid Physics, China Academy of Engineering Physics, 621900 Mianyang, China. [3] School of Physical Science and Technology, Shanghai Tech University, 201210 Shanghai, China. [4] Beijing National Laboratory for Condensed Matter Physics, Institute of Chinese Academy of Sciences, 100190 Beijing, China. [5] These authors contributed equally: Yang-Yang Lv, Jinlong Xu. ✉email: longno.2@163.com; shyao@nju.edu.cn; xiezhenda@nju.edu.cn; ybchen@nju.edu.cn

The discovery of Weyl semimetals is important in condensed matter physics; finding the Weyl fermion and exploring some interesting physics, such as the Weyl points in electronic-band structures, Fermi arc of surface states, and the chiral anomaly[1–9], are still elusive topics in particle physics. The fingerprints of Weyl semimetals consist of Weyl points with right or left chirality and corresponding "spike-like" Berry curvatures at Weyl points resulted from breaking of either time-reversal- or inversion-symmetry[10,11]. The interaction between electromagnetic waves and chiral electrons gives rise to some uniquely nonlinear optical properties, such as the photovoltaic effect in $TaIrTe_4$ and $WTe_2$, as well as second-harmonic generation and terahertz emission in TaAs[12–15]. However, high-order harmonic generation (HHG) in Weyl semimetals, an important nonlinear dynamic process of electrons under strong-field laser excitation, has not been studied previously. To date, HHG has been observed in dielectric insulators, semiconductors, Dirac semimetals and topological insulators[16–34] and has been proven to extract electronic structure and interatomic bonding information[16–34]. Regarding HHG in Weyl semimetals, we have two basic physical considerations. First, there is quite a high carrier mobility in Weyl semimetals[35,36]; for example, in $\beta$-$WP_2$ (an experimentally confirmed type-II Weyl semimetal that is due to breaking of inversion-symmetry[36,37]), the carrier mobility can be as large as $10^6$ $cm^2/(V\,s)$[36]. High mobility suggests that electrons can move significantly in the Brillouin zone, which is advantageous for HHG in solids. Second, "spike-like" Berry curvatures in Weyl semimetals may generate even-order HHG because Berry curvature was recently proposed to efficiently give rise to even-order HHG[24,29,30]. However, these considerations have not been experimentally proven in Weyl semimetals.

Considering these backgrounds, here, we study HHG in $\beta$-$WP_2$ crystals. Remarkably, we experimentally observed efficient HHG (as high as ten order), including both odd- and even-order harmonics in $\beta$-$WP_2$ pumped by a femtosecond infrared laser with fairly low intensity (~0.29 $TW/cm^2$). Furthermore, the experimental electronic-band structure and Berry curvatures of the valence and conduction bands of $\beta$-$WP_2$, retrieved from the analysis of polarization-dependent HHG, are quantitatively in agreement with calculations performed with first-principles density functional theory. Our results provide solid experimental proof for exploiting Weyl semimetals as a pragmatic material platform for VUV (vacuum ultraviolet) photonics and an all-optical and extensible solution to derive the complicated multiband electronic structure of Weyl semimetals.

## Results and discussion

**Crystal structure and composition**. $\beta$-$WP_2$ crystallizes into a noncentrosymmetric orthorhombic structure (space group of $Cmc2_1$, No. 36; $a = 3.1649$ Å, $b = 11.1159$ Å, and $c = 4.9732$ Å) (as shown in Fig. 1a)[38]. The details of crystal growth and characterizations of $\beta$-$WP_2$ crystals are summarized in the "Methods" section. Figure 1b presents the optical micrograph of $\beta$-$WP_2$ crystals grown by the chemical vapor transport (CVT) method. The stoichiometric ratio of 1:2 between W and P was determined by energy dispersive spectroscopy (EDS) (see Fig. 1c). The corresponding XRD pattern of the as-grown single crystals is shown in Fig. 1d. Obviously, only reflections of the (02$l$0) planes were detected, which suggests that the exposed surface of crystals is the $ac$-plane. In addition, the full-width-at-half-maximum of the (001) pole is as small as 0.06°, verifying that as-grown $\beta$-$WP_2$ crystals are of high crystalline quality. The carrier concentration and mobility, shown in Supplementary Fig. 1 in the supplementary information, were determined to be on the order of $10^{21}$ $cm^{-3}$ and

$10^5$ $cm^2/(V\,s)$, respectively. These properties may be advantageous for efficiently achieving HHG in $\beta$-$WP_2$.

**HHG spectra**. We used a femtosecond laser (1900 nm, 120 fs, maximum peak intensity of ~1.2 $TW/cm^2$) to explore HHG in $\beta$-$WP_2$ crystals (shown in Fig. 2a). The as-grown samples were irradiated by a linearly polarized laser emitted from an amplified Ti-sapphire laser-pumped optical parametric amplifier. The efficient yield of HHG was detected when the $a$-axis ($\Gamma$–$X$ direction in reciprocal space) of $\beta$-$WP_2$ was oriented perpendicular to the laser polarization. As shown in Fig. 2b, the measured spectrum is composed of multipeak radiations including both odd- and even-order (from 2 to 10) harmonics. The tenfold harmonics cover a broad spectrum from IR (950 nm, 1.3 eV) to VUV (190 nm, 6.5 eV). The VUV HHG response is so robust/strong that radiation could be measured in an atmospheric environment. To reveal the physical mechanism of HHG in $\beta$-$WP_2$, we studied the evolution of HHG intensity when the laser intensity was changed from 0.1 to 1.2 $TW/cm^2$ (see Fig. 2c). Figure 2d displays a false color representation of the HHG spectra versus excitation intensity. As shown in Fig. 2d, the pump threshold can be as low as 0.23 $TW/cm^2$ for 9th harmonic generation and 0.29 $TW/cm^2$ for 10th-order harmonic generation, which are both lower values than those of most previous HHG works on semiconductors/insulators (see Supplementary Table 1 and Supplementary Fig. 2). Such a low pump threshold for HHG in $\beta$-$WP_2$ may be due to its high mobility. Increasing the pump threshold to its maxima at 1.2 $TW/cm^2$, we did not observe an HHG cut-off in $\beta$-$WP_2$, which is different from the observations in ZnO and solid Ar and Kr, where two or more plateaus associated with extending HH orders were successively generated with increasing pump intensity[31,39]. This discrepancy may come from the existence of higher orders larger than 10th in $\beta$-$WP_2$. However, the detection of orders higher than 10th is restricted by the 180-nm cut-off of the commercial spectrometer used in our experiment.

Meanwhile, the radiation strength as a function of excitation intensity was studied, and the results of the 5th–10th orders are shown in Fig. 2e (data of the 2nd–4th orders are shown in Supplementary Fig. 3). Evidently, all the orders follow the approximately same power-law dependence, with exponents ranging from 2.1 to 2.5, similar to the power law observed in previous HHG studies on $MoS_2$ and graphene[30,31]. This dependence is entirely different from that of conventional perturbative nonlinear optics[40,41], which follows an $I_{ex}^N$ dependence for the Nth order on the excitation intensity $I_{ex}$. This suggests that the HHG observed in $\beta$-$WP_2$ crystals comes from Bloch oscillations rather than conventionally perturbative nonlinear processes. Note that there is significant HHG observed in Weyl semimetals, providing potential material candidates beyond semiconductors/insulators to achieve HHG.

**Theoretical mechanism of HHG**. In what follows, we try to understand the physical mechanism of the abovementioned HHG in $\beta$-$WP_2$. Currently, the physical mechanisms of HHG in solids, especially even-order HHG, are a topic of ongoing debates. For example, interband transitions and successive Bloch oscillations, sole Bloch oscillations of intraband electrons, and interband resonant high-harmonic generation have been proposed to explain HHG in solids[24,28,30,42–44]. In this work, we tentatively propose that odd-order HHG in $\beta$-$WP_2$ is attributed to intraband Bloch oscillations, while even-order HHG in $\beta$-$WP_2$ comes from the intraband Bloch oscillations under "spike-like" Berry curvature. Some qualitative discussions that rule out interband transition/resonance mechanisms and perturbative nonlinear optics

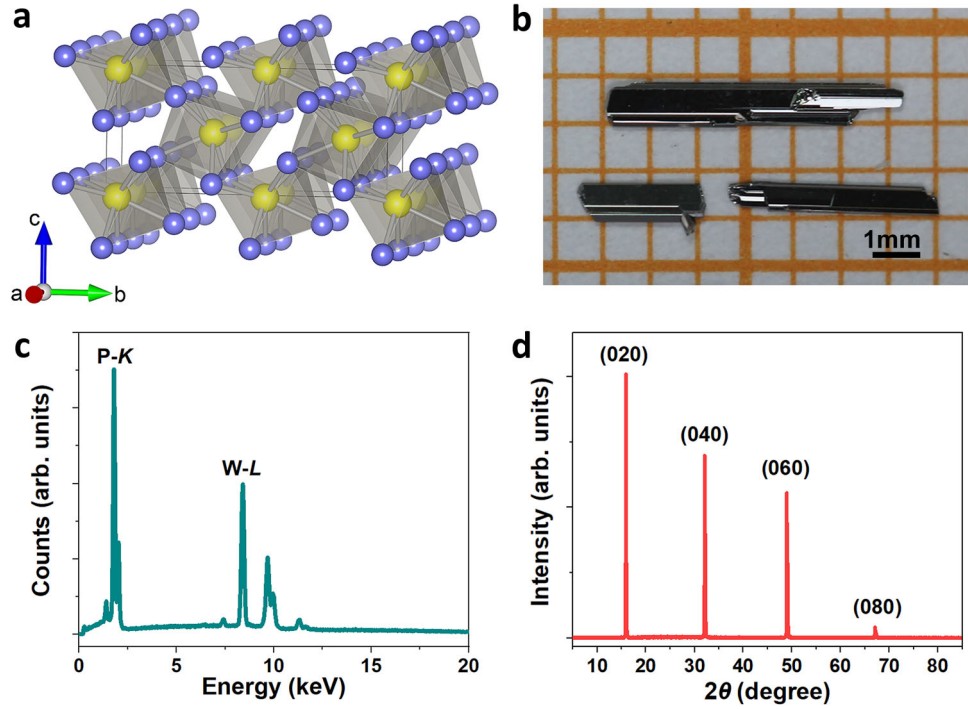

**Fig. 1 Structure and composition information of β-WP₂. a** Crystal structure of β-WP₂. **b** Typical photograph of the as-grown β-WP₂ single crystals. **c** EDS data of the β-WP₂ single crystal. **d** Corresponding XRD patterns of a representative single crystal.

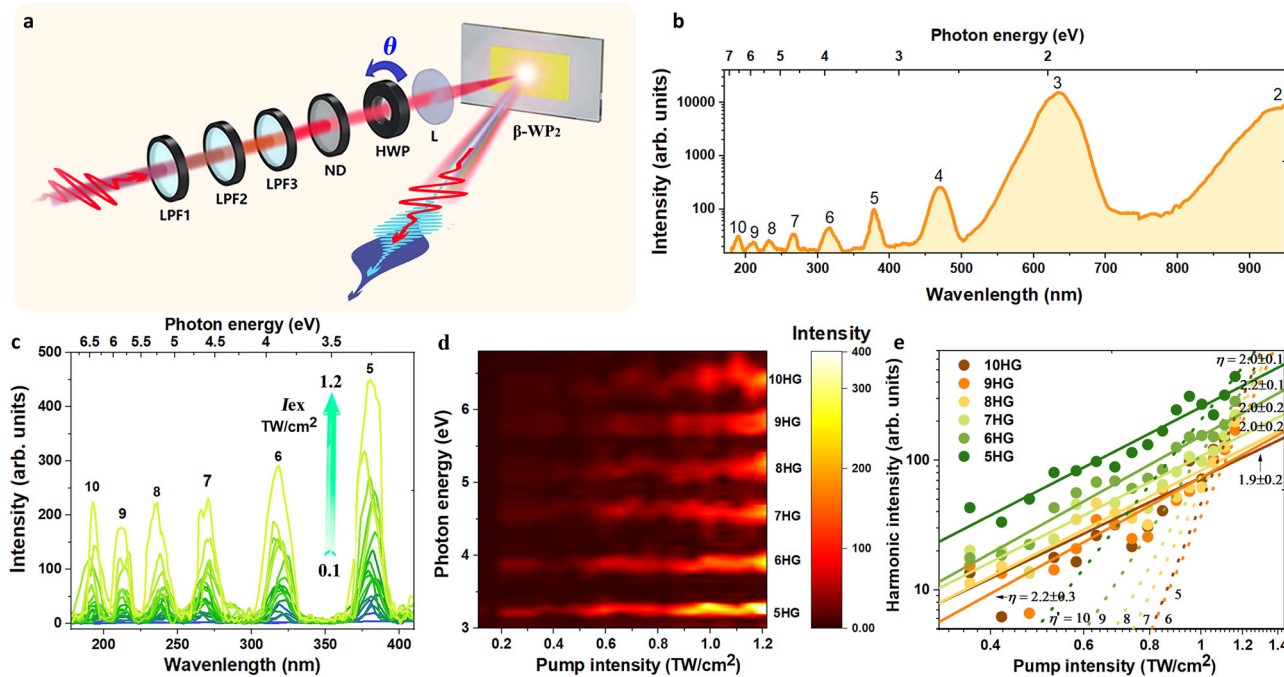

**Fig. 2 High-harmonic spectra emitted from β-WP₂ and corresponding pump intensity dependence at a carrier wavelength of 1900 nm. a** Schematic diagram of the HHG experimental apparatus. LPF1-3: a series of long-wavelength pass filters (>1500 nm); ND: continuously variable neutral density filter; HWP: half-wave plate; L: focus lens with $f = 85$ mm. The pump intensity is adjustable by rotating the attenuation of the ND. The linearly polarized electric field of the pump laser, which is originally vertical, is rotatable by controlling HWP rotation. **b** Recorded harmonic spectra emitted from β-WP₂ crystals up to the 10th order (190 nm, 6.5 eV) in the perpendicular polarization configuration. **c** High-harmonic spectra of the 5th–10th orders as a function of the pump intensity ($I_{ex}$) ranging from 0.1 to 1.2 TW/cm². All spectra are obtained with an integration time of 1 s. **d** Corresponding false-color map of the intensity dependence, represented by integrating the spectra in (**c**). **e** Logarithmic plot of the strength of the 5th–10th harmonic peaks with respect to excitation intensity. Fitting the experimental data using the power-law equation $I \propto I_{ex}^n$ gives the values of the $n$-factor for each harmonic (solid lines). The fitted $n$ of all high-order harmonics ranging from 2.1 to 2.5 suggests that the HHG observed in β-WP₂ comes from Bloch oscillations, rather than perturbative nonlinear optics. The dependence of the perturbative nonlinear optics $n = N$ of the $N$th harmonic is presented for comparison (dashed lines, fitting at the highest intensities). All the above measurements were performed at room temperature (300 K).

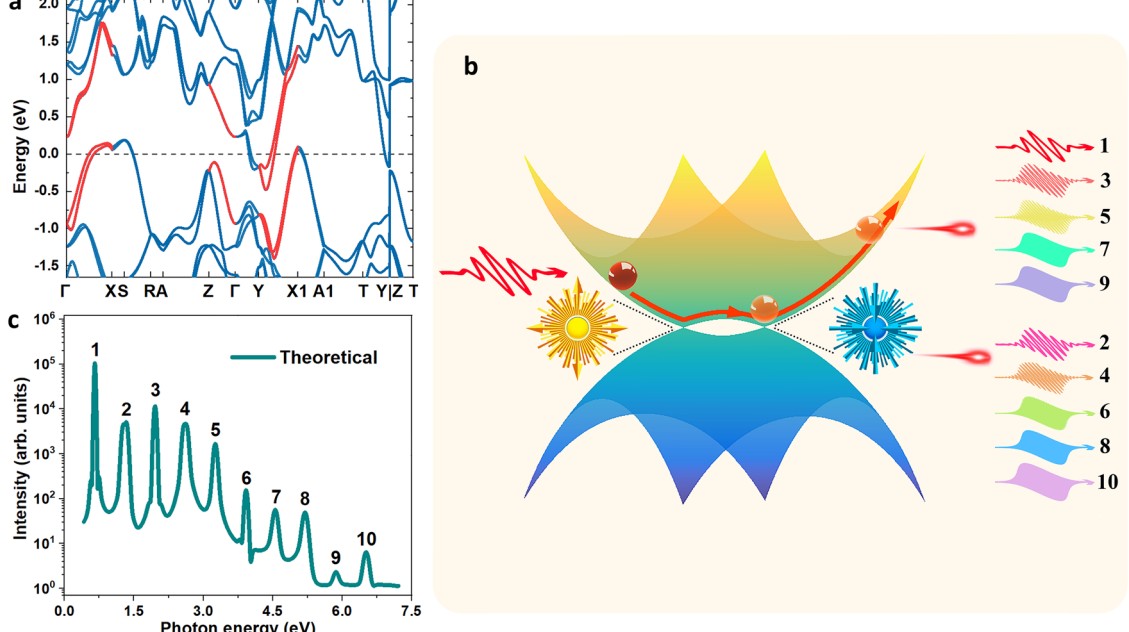

**Fig. 3 Theoretical HHG of β-WP₂. a** Electronic-band structure of β-WP₂ considering spin-orbit coupling. **b** Semiclassical picture of the mechanism for the odd- and even-order HHG generated in β-WP₂. **c** Calculated HHG spectrum of β-WP₂.

mechanisms can be found in Supplementary Note 6. Figure 3a shows the theoretical electronic-band structure of β-WP₂. In the calculation, the spin-orbit interaction was considered self-consistent. Evidently, there are two branches of electronic-band structures in both valence and conduction bands at the Fermi energy, which is the fingerprint of semimetals. At the degenerate points (Weyl points) of these two branches of electronic bands, there are "spike-like" Berry curvatures, which have been reported in previous literature[36,45]. The theoretical Berry curvatures of β-WP₂ can be seen in Supplementary Fig. 4. Figure 3b shows a schematic of odd- and even-order HHG in β-WP₂ crystals. In accordance with the intraband Bloch oscillation mechanisms, odd-order HHG comes from the nonlinear acceleration of electrons in the electronic-band structure ($E_i(k) = \sum_{n=0}^{2} \varepsilon_{i,n} \cos(nka)$), where $i$, $k$, and $a$ are the band index, the Bloch wavevector and distance between the nearest neighboring atoms, respectively. Here there are only cosine functions in $E_i$ expression because time-reversal-symmetry in β-WP₂ makes the $E_i$ being an even-function of Bloch wavevector $k$[46]. $\varepsilon_{i,n}$ is the hopping coefficient of $n$-order neighboring hopping in the $i$-th band under the stimulus of a short-pulsed (femtosecond) laser. The intensity of odd-order HHG can be written as follows:

$$I_{i,2m+1} \propto \left[(2m+1)\omega_L\right]^2 \left|\sum_n na\varepsilon_{i,n} J_{2m+1}\left[\frac{neE_0 a}{\hbar\omega_L}\right]\right|^2 \quad (1)$$

where $\omega_L$, $E_0$, $e$ and $\hbar$ are the ponderomotive frequency of the fundamental wave, amplitude of the fundamental wave, electron charge and reduced Planck constant, respectively. $J_{2m+1}$ is the $(2m+1)$-order cylindrical Bessel function. In contrast, even-order HHG is attributed to the extra "Lorentz force" coming from the Berry curvature in the dynamic equations (see the second term of Eq. (3)) of the wave pocket of the Bloch electron, which are written as follows:

$$\hbar\dot{\mathbf{k}} = -e\mathbf{E} \quad (2)$$

$$\dot{\mathbf{r}} = \frac{1}{\hbar}\frac{\partial \varepsilon(\mathbf{k})}{\partial \mathbf{k}} - \dot{\mathbf{k}} \times \boldsymbol{\Omega}(\mathbf{k}) \quad (3)$$

Based on these dynamic equations, the even-order HHG intensity can be written as follows:

$$I_{2m} \propto \left(2m\omega_L \frac{eE_0}{\hbar}\right)^2 \left|\sum_n \gamma_n \left[J_{2m-1}\left(\frac{neE_0 a}{\hbar\omega_L}\right) - J_{2m+1}\left(\frac{neE_0 a}{\hbar\omega_L}\right)\right]\right|^2 \quad (4)$$

where $\gamma_n$ is the $n$-order expansion coefficient of Berry curvature in reciprocal lattices. Berry curvature $\Omega_i$ is also the periodic function at the Brillouin zone; mathematically, $\Omega_i(k) = \sum_{n=0}^{2} \gamma_{i,n} \sin(nka)$. Here there are only sine functions in $\Omega_i$ expression because time-reversal-symmetry makes the $\Omega_i$ being an odd-function of Bloch wavevector $k$[46]. The derivation of the above formula can be found in Supplementary Note 8. Figure 3c shows the theoretical HHG of β-WP₂ calculated by Eqs. (1) and (4) with the stimulation of a 1900 nm infrared femtosecond laser. Evidently, β-WP₂ exhibits both even- and odd-order HHG, which agrees with the experimental results. In addition, according to Eqs. (1) and (4), both odd- and even-order harmonics show a similar power-law dependence on the intensity of the fundamental wave (see Supplementary Note 8). Certainly, the above analysis is based on a simplified tight-bonding model, and we wish theoretical readers to directly calculate HHG in β-WP₂ by the time-dependent first-principles density functional theory[47]. One thing should be emphasized that the breaking-inversion-symmetry in β-WP₂ plays the crucial role on observed even-order HHG. Because breaking-inversion-symmetry gives rise to the Weyl semimetal state in β-WP₂ and accordingly the "spike-like" Berry curvature at Weyl points[10], in turn, there is significant even-order HHG in β-WP₂.

**Polarization-dependent HHG.** To further identify the HHG characteristics, the evolution of the HHG spectra with different crystallographic orientations was measured by rotating the linear polarization angle $\theta$ of the pump to change the angle between the polarization direction and the $a$-axis (see Fig. 2). This was performed by carefully controlling the linear polarization angle of the pump by a half-wave plate to adjust the angle $\theta$ from 0 to $2\pi$. Figure 4a, b presents the spectral maps of the 4th–10th-order HH

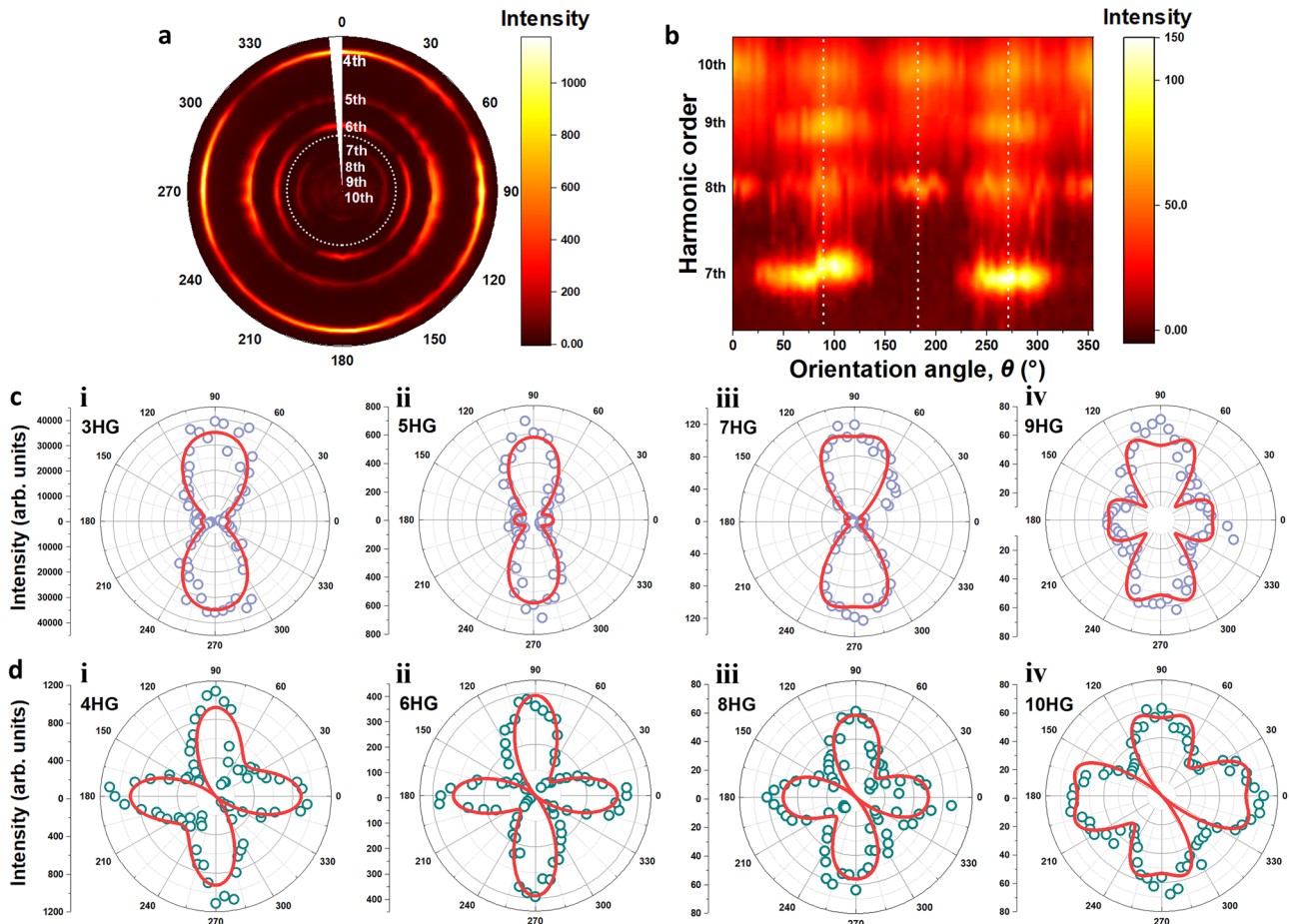

**Fig. 4 Dependence of the high-harmonic yield from β-WP₂ on pump orientation. a** False-color map of the crystallographic orientation-dependent HHG spectra. Zero degrees corresponds to the pump polarization parallel to the *a*-axis (*Γ–X* direction in reciprocal space). The pump intensity was set as ~0.9 TW/cm² at a wavelength of 1900 nm in the experiment. **b** Zoomed-in view of the section of the 7th–10th harmonics surrounded by dashed circles in (**a**). Vertical lines highlight angles of 90°, 180°, and 270°. **c**, **d** Polar plots of the respective orientation-dependent strength of odd and even orders. Red lines are the fitting results obtained from Eqs. (5) and (6).

spectra and the corresponding magnified 7th–10th-order HH spectra, respectively. To avoid damaging the sample during long-term irradiation, the pump intensity of the 1900-nm femtosecond laser was limited to ~0.9 TW/cm² in the experiment. Figure 4c, d demonstrates the remarkably different polarization dependences of odd- and even-order HHG because odd- and even-order HHG responses are generated from different mechanisms, as shown in Eqs. (1) and (4). The even orders can reach their maxima when the polarization is parallel ($\theta = 0°, 180°$) or perpendicular ($\theta = 90°, 270°$) to the *a*-axis, but the odd orders attain maxima only when the polarization is perpendicular to the *a*-axis. As shown in Fig. 4c, the experimentally observed polar map of odd-order HHG can be well fitted by considering Bloch oscillations of electrons along two orthogonal directions of electronic dispersion. Mathematically, the fitting criterion $C_{\text{band}}$ is defined as follows:

$$c_{\text{band}} = min\left\{\sum_{\theta_i}\left[I_{\text{theory}}^{\text{band}}(\theta_i, t_1, t_2, t_3, t_4) - I_{\text{exp}}^{\text{odd}}(\theta_i)\right]^2\right\} \quad (5)$$

where $I_{\text{theory}}^{\text{band}}(\theta_i, t_1, t_2, t_3, t_4)$ is the theoretical intensity of odd-order HHG considering the hopping coefficients ($t_1, t_2, t_3,$ and $t_4$) of the electronic-band structure under tight-binding approximation conditions. $I_{\text{exp}}^{\text{odd}}(\theta_i)$ is the experimentally measured intensity of odd-order HHG in β-WP₂ crystals. The mathematical expression of $I_{\text{theory}}^{\text{band}}(\theta_i, t_1, t_2, t_3, t_4)$ can be found in

Supplementary Note 8 (Supplementary Eq. (S15)). As shown in Fig. 4, the experimental θ-dependent HHG in β-WP₂ crystals can be well fitted by Eq. (5). In the simulation, we only considered the two hopping coefficients (the nearest and next-nearest neighboring hopping) because using more parameters results in too many possibilities. Note that we have considered the Bloch oscillations of electrons in both the conduction and valence bands around the Fermi energy in β-WP₂. The comparisons of the experimental and theoretical electronic-band structures are shown in Fig. 5.

Similar to odd-order HHG, as shown in Fig. 4d, the experimental even-order HHG of β-WP₂ can also be well fitted by theoretical results by considering the effect of the Berry curvature of β-WP₂, as we discussed in Fig. 3b and c. Similar to extracting the electronic-band structure (Eq. (5)), the fitting criterion $C_{\text{Berry}}$ can be written as follows:

$$c_{\text{Berry}} = min\left\{\sum_{\theta_i}\left[I_{\text{theory}}^{\text{Berry}}(\theta_i, \gamma_1, \gamma_2, \gamma_3, \gamma_4) - I_{\text{exp}}^{\text{even}}(\theta_i)\right]^2\right\} \quad (6)$$

where $I_{\text{theory}}^{\text{Berry}}(\theta_i, \gamma_1, \gamma_2, \gamma_3, \gamma_4)$ is the theoretical intensity of even-order HHG considering the expansion coefficients ($\gamma_1, \gamma_2, \gamma_3,$ and $\gamma_4$) of the Berry curvature (see Eq. (S16) in Supplementary Note 8). $I_{\text{exp}}^{\text{even}}(\theta_i)$ is the experimentally measured intensity of even-order HHG in β-WP₂ crystals. One can see from Fig. 4d that

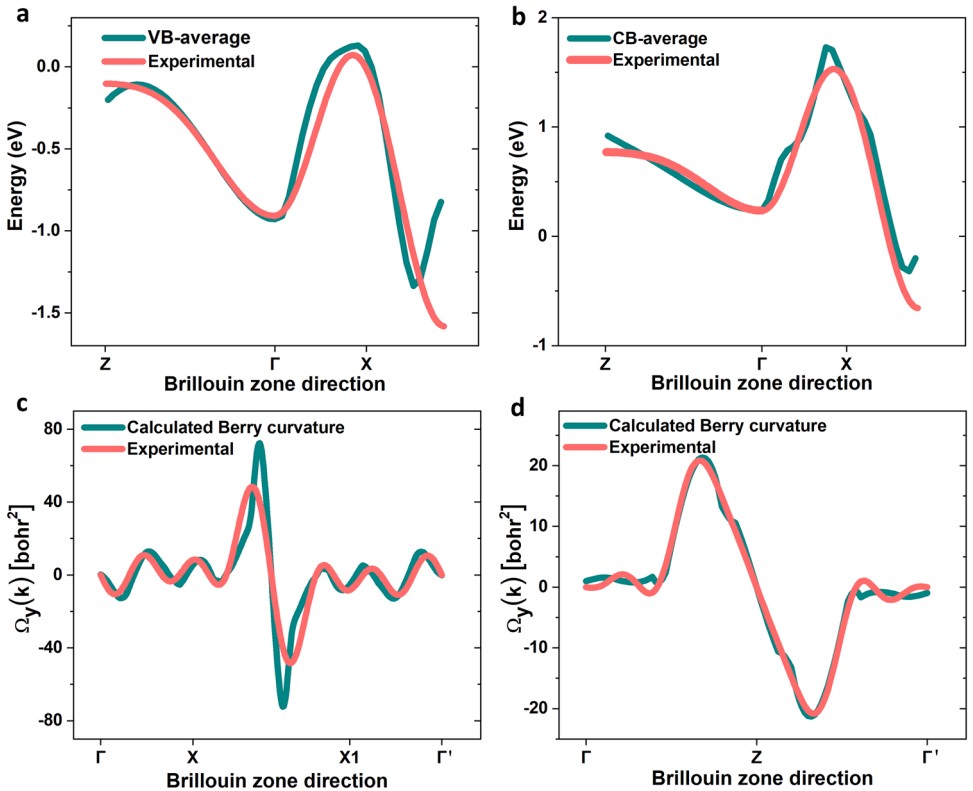

**Fig. 5 Theoretical and experimental electronic structure and Berry curvature of β-WP₂. a, b** The blue lines are the electronic dispersion curves of the conduction and valence bands crossing the Fermi energy ($E_F = 0$) obtained by first-principles calculation. The red lines are the electronic dispersion curve retrieved by fitting the $\theta$-dependent odd-order HHG of β-WP₂ crystals. The two results fit well. **c, d** The blue lines are the theoretically calculated Berry curvature of β-WP₂. The red lines are the experimental Berry curvature of β-WP₂ fitted by even-order HHG, which is in good agreement with the theoretical curves.

$I_{\exp}^{\text{even}}(\theta_i)$ is well fitted by even-order HHG generated by the Berry curvature.

**Electronic structure and Berry curvature.** Theoretically, the fitting parameters of the $\theta$-dependent odd-order HHG of β-WP₂ are hopping coefficients of the electronic-band structure in β-WP₂. In Fig. 5a, b, we compare the electronic-band structures of conduction and valence bands crossing the Fermi energy ($E_F = 0$) obtained by first-principles density functional theory and the experimental results retrieved through fitting $\theta$-dependent odd-order HHG. Evidently, the experimental electronic-band structures are in agreement with the theoretical structures. Note that in our experiment, the electrical field of the femtosecond laser is oriented along the $ac$-plane of β-WP₂ crystals; accordingly, we can only extract the electronic band along high-symmetric directions (highlighted by red in Fig. 3a) at the $ac$-plane of the Brillouin zone. This vividly demonstrates that HHG can be an effective method to study the electronic-band structures of metals/semimetals; previously, its effectiveness was only demonstrated in semiconductors/insulators. Note that angle-resolved photoemission spectroscopy (ARPES) is the most commonly used method to characterize the electronic-band structure of solids. However, compared with ARPES, HHG is a bulk-sensitive method that effectively prevents the surface contamination and surface reconstruction artifacts universally observed in ARPES. Therefore, HHG may be an effectively alternative method complementary to ARPES that can be used to study the electronic-band structures of solids.

Similar to the case of the electronic-band structure, analysis of even-order HHG can provide information on the Berry curvatures of β-WP₂, which are depicted in Fig. 5c, d. Obviously, the experimental Berry curvatures of the valence band and conduction band of β-WP₂ are also in agreement with the theoretical curvatures. This vividly demonstrates the usefulness of HHG to extract the Berry curvature of quantum topological materials.

In conclusion, we experimentally observed high-harmonic generation as high as ten order, extending into the VUV region and including both odd- and even-order generation, in type-II Weyl semimetal β-WP₂ crystals under femtosecond infrared laser irradiation with a fairly low intensity. The experimental electronic-band structure and Berry curvature around the Fermi energy of β-WP₂, retrieved through the analysis of polarization-dependent HHG, are in good agreement with the theoretical electronic-band structure and Berry curvature calculated by first-principles density functional theory. Our work demonstrates the promising applications of Weyl semimetals in HHG, such as the generation of ultraviolet laser sources, and HHG is an alternative/complementary method that can be used to experimentally retrieve the complicated electronic-band structure and Berry curvature of quantum topological materials.

## Methods
**Crystal growth.** The single crystals of β-WP₂ were prepared by a chemical vapor transport (CVT) technique. In the growth procedure, high purity red phosphorous (P, Alfa-Aesar, 99.999%) and tungsten (W, Alfa-Aesar, 99.999%) were used as the starting materials. They were mixed and sealed in an evacuated quartz tube with 10 mg/cm³ iodine (I₂, Alfa-Aesar, 99.9985%) as a transport agent, and put into a two-zone-furnace with a temperature gradient of 1000 °C to 900 °C to grow crystals. After over 10 days, the millimeter-sized needle like β-WP₂ single crystals with metallic-luster were obtained.

**Crystal characterizations**. The element analysis was performed using an energy dispersive spectroscopy (EDS) spectrometer equipped in a scanning electron microscope (SEM) (FEI-Quanta). X-ray diffraction (XRD) measurement on an X-ray diffractometer (Ultima III Rigaku) using Cu-$K_\alpha$ radiation with $2\theta$ of 5~85° was carried out to determine the surface orientation of the β-$WP_2$ single crystals. Longitudinal resistivity $\rho_{xx}$ and Hall (transversal) resistivity $\rho_{xy}$ were measured by a standard four-probe configuration on a 9 T Quantum Design Physical Property Measurement System (PPMS). The dielectric constant and refractive index of β-$WP_2$ were directly measured by a variable-angle spectroscopic ellipsometer.

**HHG experiments**. The HHG data presented in this work were taken in a reflection spectroscopy geometry. A Ti-sapphire chirp-pulse amplifier system was used as femtosecond laser source, producing 800 nm, ~120 fs, 0.7 mJ pulses at a repetition rate of 5 kHz. The output beam was employed to pump an optical parametric amplifier (OPA), with idler wavelength set at 1.9 μm, with respect to a signal wavelength of 1.4 μm. A series of long-pass filters (>1500 nm) were utilized to ensure an adequate filtering of signal beam and shorter wavelengths accompanied with idler beam emitted from OPA through difference frequency generation (DFG). The as-grown β-$WP_2$ single crystal, mounted on a $SiO_2$ plate, is oriented with a-axis along the horizontal direction. With a concave $CaF_2$ lens of 85-mm focal length, the 1.9-μm pump beam was focused onto β-$WP_2$ specimen with a small incident angle (~5°). The maximum pump intensity of 1.2 $TW/cm^2$ is close to the observed damage threshold of β-$WP_2$, possibly due to thermal accumulation enhanced by high-density surface-electron relaxation. The HHG spectra reflected off β-$WP_2$ were detected and recorded by a commercial spectrometer (OceanOptics, wavelength range of 180–1100 nm) coupled with an ultraviolet fiber, for either intensity- or polarization-dependence measurements. The fiber port was fixed as closed as possible to β-$WP_2$ with an acceptance angel of ~9°. The whole experimental apparatus was processed in air environment and room temperature.

**Theoretical calculations**. All the density functional calculations in this work are implemented in the Vienna Ab-initio Simulation Package (VASP) code[48,49]. The projected augmented wave method[50,51] and the generalized gradient approximation with the Perdew–Burke–Ernzerhof exchange-correlation[52] are used. In the calculations, we use the plane-wave cut-off energy of 350 eV and a k-mesh of $11 \times 11 \times 13$. The spin-orbit coupling (SOC) effect is included in all the calculations. The Berry curvature is calculated assistant by the Wannier90 code[53].

## Data availability
The data collected or analyzed during this study are found in the main text or the supplementary information. All raw data related to the current work are available from the corresponding authors upon reasonable request.

## Code availability
Details on the numerics are available upon request from the authors.

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

## Acknowledgements

The authors acknowledge financial support from the National Key R&D Program of China (2017YFA0303700, 2019YFA0705000), the State Key Program for Basic Research of China (973 Program) (2015CB659400), the National Natural Science Foundation of China (11574131, 11774161, 51902152, 51872134, 11890702, 11690031, 11627810, and 11674169), the major research program of the National Natural Science Foundation of China (51890861), the Foundation for Innovative Research Groups of the National Natural Science Foundation of China (51721001), the Fundamental Research Funds for the Central Universities (14380157), and the National Key R&D Program of China (2016YFA0201104). Y.-Y. Lv acknowledges financial support from the Innovation Program for the Talents of China Postdoctoral Science Foundation (BX20180137) and support from the China Postdoctoral Science Foundation (2019M650105). Y.B. Chen thanks Prof. D. Sun at Beijing University for enlightening discussion. Y.B. Chen thanks Prof. D. Sun at Beijing University for enlightening discussion.

## Author contributions

Y.B.C. conceived the study. Y.-Y.L. prepared the crystal samples and made the material characterizations with the assistance of S.-H.Y. J.L.X. carried out the HHG measurements supervised by Z.D.X and S.N.Z. Y.-Y.L. analyzed the data in assist of S.H. and C.Z. Y.D.H. performed TA measurements under the guidance of J.H. J.Z. did the density functional theory calculations. Y.-Y.L., J.L.X., Y.B.C. and Y.-F.C. wrote the manuscript assisted by S.-H.Y. Z.D.X., X.-P.L., M.-H.L., H.M.W. and S.N.Z. revised the manuscript. All authors contributed to the discussions.

## Competing interests

The authors declare no competing interests.
