## [Peer Review File · Nature Communications]

REVIEWER COMMENTS

Reviewer #1 (Remarks to the Author):

Lv, et al. reported the observation of both even- and odd-order the high harmonic generation in Weyl semimetal β -WP₂ crystals. The highest order of harmonic generation can be as large as ten. Through analysis of polarization dependent intensity of harmonic generation, they concluded that the even-order harmonic generations are attributed to significant Berry curvature; while the odd ones come from the Bloch oscillations.

The discovery of Weyl semimetals is one of most important progresses in condensed matter physics within past 10 years. Up to now, the electronic band structures of Weyl points, as well as Fermi arc of electronic surface state, have been verified experimentally. Currently, this field faces an important question: can we find any useful functionality of these Weyl semimetals, except these remarkably electronic features? Lv et al. reported the efficient the high harmonic generation (up to ultra-violet regime) in Weyl semimetal β -WP₂ crystals. According to my knowledge, this draft reports the first experimental observation of the high-harmonic-generations (HHG) in Weyl semimetal metals.

Considering this background, I think this draft has two bright spots. Firstly, for HHG, this work dramatically expanded the materials candidate besides the normal semiconductors. The Weyl semimetals were proposed by the authors to generate uniform HHG due to the superior advantages including extremely high carrier mobility and “spike-like” Berry curvatures. Secondly, they can retrieve the electronic band structure and Berry curvature of multi-bands Weyl semimetals β -WP₂ crystals through HHG experiments. This work demonstrates the HHG as a complementary method for ARPES to study the electronic band structure of topological materials. Therefore, I believe that this work can attract broad interests in the physics, optics, and materials science communities. Therefore, in my opinion, this paper is timely and systematic, and deserves the publication in Nature Communications.

But, before I finally suggest accepting this draft, I wish authors clarify/correct four important issues described as follows.

(1) In the simulation of the experimental θ -dependent HHG in β -WP₂ crystals, why the authors just considered the two hopping coefficients (the nearest and next-nearest neighboring hopping). I wish authors make more discussion about this issue.

(2) Recently Nature Physics published a paper [Bai, Y. et al. Nat. Phys. (2020). <https://doi.org/10.1038/s41567-020-01052-8>] that reported the observation of HHG in topological insulator BiSbTeSe₂. Authors should compare their results with this Nature Physics paper, and discuss the similarities and differences between these two works.

(3) There are some theoretical and experimental works reporting the odd-order HHG in Dirac semimetals, for example Cd₃As₂, (1. Kovalev, S. et al. Non-perturbative terahertz high-harmonic generation in the three-dimensional Dirac semimetal Cd₃As₂. Nat. Commun. 11, 2451 (2020); 2. Lim, J. et al. Efficient generation of extreme terahertz harmonics in three-dimensional Dirac semimetals. Phys. Rev. Research 2, 043252 (2020).). Authors should add them in the introduction to respect these works of HHG in topological materials. Both Dirac and Weyl semimetals may be good candidates to generate ultraviolet laser beam.

(4) English in this draft should be polished carefully. There are some obvious typo errors in the current version.

Reviewer #2 (Remarks to the Author):

Lv et al. report an interesting study of high-order harmonic generation (HHG) in semimetal β -WP₂ crystals. There are two important results: First, they observe

polarization-dependent HHG in a type-II Weyl semimetal. Second, they demonstrate how the polarization dependence of odd- and even-order harmonics can be used to extract the electronic band structure and Berry curvature around the Fermi-energy of the crystal. These are two major results, both of which are interesting and intriguing, however, a number of conceptual and minor issues should be addressed as discussed below.

1. I highly encourage the authors to check the English. Some writing is not correct and even confusing. I recommend a full manuscript language editing by a native English speaker, checking for typos, grammar etc..

2. The introduction suffers from exaggerating language, e.g. (*Discovery of Weyl semimetals is one of the most important progress in physics ... , The most remarkable features of Weyl semimetals are existences of Weyl points ... , ... we believe that HHG is an ideal tool ...*). Keep in mind that the introduction is not just a sales pitch. The reader may not want to know what you consider as most remarkable, or what you believe in. In addition, I would recommend adding one or two sentences stating a clear research question at the end of the introduction paragraph. What is the goal? It is clear that it HHG in a Weyl semimetal hasn't been reported before, but the manuscript would strongly benefit from a more scientific motivation.

3. While I find the overall study exciting, the manuscript needs improving for consistency. In particular, I would recommend introducing the experimental data before presenting the theoretical analysis of the even- and odd-order harmonics (which is hard to follow without reading the entire SI). The authors could achieve this by moving the theory paragraph (line 114-138) below the experimental paragraph. My recommendation is to have the experimental data stand for itself, independent of the specific interpretation, where the odd-order harmonics are attributed to Bloch oscillations, and the even-harmonics are interpreted in terms of Berry curvature effects.

4. In Figure 2e), you plot the HHG intensity versus pump intensity. Can you include uncertainties for the extracted fitting exponents?

5. For the discussion and theory results, I would have appreciated some more literature citations reflecting the open questions and controversial discussions regarding the physical mechanisms underlying HHG in solids. This deficiency makes it difficult to evaluate the validity of the model, seemingly matching the experimental data on a quantitative level. In addition, could you further elaborate why both odd- and even-order harmonics show a quadratic intensity-dependence despite the seemingly different underlying physical mechanisms? Could this hint to a common thermal origin?

6. In Figure 4, the red lines retrieved through fitting the experimental data show jumps at the gamma points. Can you discuss this feature? Is it a fitting artifact?

7. In the supplement, the x-axis in Figure S3 d) is incorrectly labeled as $B(T)$, instead of $T(K)$. Overall, I believe this is a solid and interesting study that both established unambiguously the possibility of HHG in Weyl semimetals, and also uncovers interesting puzzles for future work concerning the nature of HHG in topological materials. Assuming that the minor issues and questions raised can be addressed, I recommend this manuscript for publication.

Reviewer #3 (Remarks to the Author):

The manuscript reports on a study of near-infrared/optical high-harmonic generation in the Weyl semimetal WP2. Even- and odd-order harmonic radiation up to the 10th orders is observed by exciting the material by 1900 nm laser pulses. While the observed odd harmonics are ascribed to Bloch oscillations, the even harmonics are interpreted as a result of Berry curvatures which is characteristic for a Weyl semimetal. Nonlinear optical effects are in general a very interesting topic, thus the present study fits the scope of

this journal. However, the interpretation of the experimental results has clear flaws.

1. For many solid-state materials the observation of only odd-order harmonics (see Table S1) is guaranteed by the existence of centrosymmetry in the crystal structure. Here, WP2 crystalizes in a non-centrosymmetric structure, thus it is natural to observe the even harmonics, even without invoking the effects of Berry curvatures.

2. The band structure exhibits several bands crossing the Fermi surface, but only selected regions are compared to the experimental results as shown in Fig.4.

3. Since the energy of the laser pulses is quite high and allows interband transitions, even more bands can be involved in the dynamical processes, which however is not discussed in the manuscript.

With these considerations I cannot recommend the present manuscript for publication in Nature Communications.

Response Letter to the Reviewers

Reviewer #1 (Remarks to the Author):

Lv, et al. reported the observation of both even- and odd-order the high harmonic generation in Weyl semimetal β -WP₂ crystals. The highest order of harmonic generation can be as large as ten. Through analysis of polarization dependent intensity of harmonic generation, they concluded that the even-order harmonic generations are attributed to significant Berry curvature; while the odd ones come from the Bloch oscillations.

The discovery of Weyl semimetals is one of most important progresses in condensed matter physics within past 10 years. Up to now, the electronic band structures of Weyl points, as well as Fermi arc of electronic surface state, have been verified experimentally. Currently, this field faces an important question: can we find any useful functionality of these Weyl semimetals, except these remarkably electronic features? Lv et al. reported the efficient the high harmonic generation (up to ultra-violet regime) in Weyl semimetal β -WP₂ crystals. According to my knowledge, this draft reports the first experimental observation of the high-harmonic-generations (HHG) in Weyl semimetal metals.

Considering this background, I think this draft has two bright spots. Firstly, for HHG, this work dramatically expanded the materials candidate besides the normal semiconductors. The Weyl semimetals were proposed by the authors to generate uniform HHG due to the superior advantages including extremely high carrier mobility and “spike-like” Berry curvatures. Secondly, they can retrieve the electronic band structure and Berry curvature of multi-bands Weyl semimetals β -WP₂ crystals through HHG experiments. This work demonstrates the HHG as a complementary method for ARPES to study the electronic band structure of topological materials. Therefore, I believe that this work can attract broad interests in the physics, optics, and materials science communities. Therefore, in my opinion, this paper is timely and systematic, and deserves the publication in Nature Communications.

But, before I finally suggest accepting this draft, I wish authors clarify/correct four important issues described as follows.

Answer: Firstly, we'd like to thank you for your review time and your constructive comments on our draft. We carefully read your comments and re-analyze the original data in accordance with your suggestions. The responses to your comments are outlined as follows. The corresponding revises at the main text are labeled by red color.

(1) In the simulation of the experimental θ -dependent HHG in β -WP₂ crystals, why the authors just considered the two hopping coefficients (the nearest and next-nearest neighboring hopping). I wish authors make more discussion about this issue.

Answer: Thanks for the reviewer's comments. In the simulation, we just considered the two hopping coefficients. There are two reasons: (1) the HHG is predominantly contributed by the nearest and next-nearest neighboring hopping; (2) considering more than two coefficients would lead to the uncertainty of the fitting results due to the more curve-fitting parameters.

(2) Recently Nature Physics published a paper [Bai, Y. et al. Nat. Phys. (2020). <https://doi.org/10.1038/s41567-020-01052-8>] that reported the observation of HHG in topological insulator BiSbTeSe₂. Authors should compare their results with this Nature Physics paper, and discuss the similarities and differences between these two works.

Answer: Thanks for the reviewer's good suggestion. Bai, Y. et al. reported the observation of HHG in topological insulator BiSbTeSe₂. They claimed that even- and odd-order harmonics have different origins, and the even-order harmonics that are produced from a freshly cleaved BiSbTeSe₂ surface and the odd-order harmonics origin in the bulk. The even-order HHG can be the fingerprints of strong-field-driven helical Dirac fermions in the topological surface states. In our paper, we discovered effective HHG in type-II Weyl semimetal β -WP₂ crystals under fairly low femtosecond laser

intensity, where the odd-order harmonics come from the Bloch electron oscillation, while the even orders were attributed to “spike-like” Berry curvature at Weyl points. Although the two papers all reported the observation of HHG in topological materials, they have many differences (especially, the one is topological insulator, and the other Weyl semimetal). Now the paper of Bai, Y. et al. is cited and discussed in our new draft.

(3) There are some theoretical and experimental works reporting the odd-order HHG in Dirac semimetals, for example Cd_3As_2 , (1. Kovalev, S. et al. Non-perturbative terahertz high-harmonic generation in the three-dimensional Dirac semimetal Cd_3As_2 . *Nat. Commun.* 11, 2451 (2020); 2. Lim, J. et al. Efficient generation of extreme terahertz harmonics in three-dimensional Dirac semimetals. *Phys. Rev. Research* 2, 043252 (2020).). Authors should add them in the introduction to respect these works of HHG in topological materials. Both Dirac and Weyl semimetals may be good candidates to generate ultraviolet laser beam.

Answer: Thanks for the reviewer’s good suggestion. This problem is our negligence. In the new draft, we cited and discussed the two papers of the odd-order HHG in Dirac semimetals [1. Kovalev, S. et al. Non-perturbative terahertz high-harmonic generation in the three-dimensional Dirac semimetal Cd_3As_2 . *Nat. Commun.* 11, 2451 (2020); 2. Lim, J. et al. Efficient generation of extreme terahertz harmonics in three-dimensional Dirac semimetals. *Phys. Rev. Research* 2, 043252 (2020).] in the introduction.

(4) English in this draft should be polished carefully. There are some obvious typo errors in the current version.

Answer: We have carefully read the whole draft and corrected some typos. The corrected languages parts have been labeled by red color.

We hope that our replies could answer your doubts. Thank you again for your careful review on our work.

Reviewer #2 (Remarks to the Author):

Lv et al. report an interesting study of high-order harmonic generation (HHG) in semimetal β -WP₂ crystals. There are two important results: First, they observe polarization-dependent HHG in a type-II Weyl semimetal. Second, they demonstrate how the polarization dependence of odd- and even-order harmonics can be used to extract the electronic band structure and Berry curvature around the Fermi-energy of the crystal. These are two major results, both of which are interesting and intriguing, however, a number of conceptual and minor issues should be addressed as discussed below.

Answer: We'd like to thank you for your review time and your constructive comments on how to make our draft more solid and clear. Your comments are addressed one-by-one as follows.

1. I highly encourage the authors to check the English. Some writing is not correct and even confusing. I recommend a full manuscript language editing by a native English speaker, checking for typos, grammar etc..

Answer: We are sorry for the problem in our old manuscript. We have revised the whole manuscript and carefully proof-read the manuscript to minimize typographical, grammatical, and bibliographical errors. In addition, we have invited a native English speaker to check the language. We believe that the language is now acceptable for the review process.

2. The introduction suffers from exaggerating language, e.g. (*Discovery of Weyl semimetals is one of the most important progress in physics ...*, *The most remarkable features of Weyl semimetals are existences of Weyl points ...*, *... we believe that HHG*

is an ideal tool ...'. Keep in mind that the introduction is not just a sales pitch. The reader may not want to know what you consider as most remarkable, or what you believe in. In addition, I would recommend adding one or two sentences stating a clear research question at the end of the introduction paragraph. What is the goal? It is clear that it HHG in a Weyl semimetal hasn't been reported before, but the manuscript would strongly benefit from a more scientific motivation.

Answer: Thanks for reviewer's good suggestions. We sincerely accepted your suggestions. In the new draft, we have modified the introduction and given a more scientific motivation to our present study. For your convenience, we copied new introduction as follows.

The discovery of Weyl semimetals is important in condensed matter physics; finding the Weyl fermion and exploring some interesting physics, such as the Weyl points in electronic band structures, Fermi arc of surface states, and the chiral anomaly [1-9], are still elusive topics in particle physics. The fingerprints of Weyl semimetals consist of Weyl points with right or left chirality and corresponding "spike-like" Berry curvatures at Weyl points [10,11]. The interaction between electromagnetic waves and chiral electrons gives rise to some uniquely nonlinear optical properties, such as the photovoltaic effect in TaIrTe₄ and WTe₂, as well as second-harmonic generation and terahertz emission in TaAs [12-15]. However, high-order harmonic generation (HHG) in Weyl semimetals, an important nonlinear dynamic process of electrons under strong-field laser excitation, has not been studied previously. To date, HHG has been observed in dielectric insulators, semiconductors, metals and topological insulators [16-32] and has been proven to extract electronic structure and interatomic bonding information [16-32]. Regarding HHG in Weyl semimetals, we have two basic physical considerations. First, there is quite a high carrier mobility in Weyl semimetals [33,34]; for example, in β -WP₂ (an experimentally confirmed type-II Weyl semimetal), the carrier mobility can be as large as 10^6 cm²/(V·s) [34]. High mobility suggests that electrons can move significantly in the Brillouin zone, which is advantageous for HHG in solids. Second, "spike-like" Berry curvatures in Weyl semimetals may generate even-

order HHG because Berry curvature was recently proposed to efficiently give rise to even-order HHG [24,29,30]. However, these considerations have not been *experimentally* proven in Weyl semimetals.

2. While I find the overall study exciting, the manuscript needs improving for consistency. In particular, I would recommend introducing the experimental data before presenting the theoretical analysis of the even- and odd-order harmonics (which is hard to follow without reading the entire SI). The authors could achieve this by moving the theory paragraph (line 114-138) below the experimental paragraph. My recommendation is to have the experimental data stand for itself, independent of the specific interpretation, where the odd-order harmonics are attributed to Bloch oscillations, and the even-harmonics are interpreted in terms of Berry curvature effects.

Answer: Thanks for reviewer's good suggestions. We are sorry for the problem in our old manuscript. According to the suggestions, we have moved the theory paragraph below the experimental paragraph. And the corresponding figures were recombined in the new draft. These changes will not influence the content of the paper.

3. In Figure 2e), you plot the HHG intensity versus pump intensity. Can you include uncertainties for the extracted fitting exponents?

Answer: Thanks for reviewer's comments. Now, the errors of the extracted fitting exponents were added in Figure 2e (see Fig. A1).

Fig. A1 Logarithm plot of the strength for 5th-10th harmonic peaks with respect to excitation intensity.

4. For the discussion and theory results, I would have appreciated some more literature citations reflecting the open questions and controversial discussions regarding the physical mechanisms underlying HHG in solids. This deficiency makes it difficult to evaluate the validity of the model, seemingly matching the experimental data on a quantitative level. In addition, could you further elaborate why both odd- and even-order harmonics show a quadratic intensity-dependence despite the seemingly different underlying physical mechanisms? Could this hint to a common thermal origin?

Answer: We admit that we have not carefully discussed this issue at the old draft.

At the revised draft, we have added below sentences to discuss the open questions and controversial discussions regarding the physical mechanisms of HHG in solids, as well as more data and analysis to rule out some other possibilities at supplementary information. For your convenience, we copied it here:

In what follows, we try to understand the physical mechanism of the abovementioned HHG in β -WP₂. Currently, the physical mechanisms of HHG in solids, especially even-order HHG, are still unknown. For example, interband transitions and successive Bloch oscillations, sole Bloch oscillations of intraband electrons, and

interband resonant high-harmonic generation have been proposed to explain HHG in solids [24, 28-30, 39-41]. In this work, we tentatively propose that even-order HHG in β -WP₂ comes from the Berry curvature mechanism, while odd-order HHG is attributed to intraband Bloch oscillations. Some qualitative discussions that rule out interband transition/resonance mechanisms and perturbative nonlinear optics mechanisms can be found in the SI.

The third reviewer also raised a similar comment, especially on the mechanism of even-order HHG in β -WP₂. At what follows, we copied the answer here for your reference.

According to our literature review, the generation of even-order harmonics is proposed by two mechanisms: **1)** the interband transition and successive the Bloch oscillation in conduction band, for example the even-order harmonic generation in GaSe [Schubert, O. et al. *Nat. Photon.* 8, 119-123 (2014); Kaneshima, K. et al. *Phys. Rev. Lett.* 120, 243903 (2018).]; **2)** the Berry curvature effect in the electronic band structure [Ghimire, S. et al. *Nat. Phys.* 7, 138-141 (2011); Liu, H. Z. et al. *Nat. Phys.* 13, 262-265 (2017); Luu, T. T. & Wörner, H. J. *Nat. Commun.* 9, 916 (2018).].

In our discussion, we took the Berry curvature mechanism. There are two reasons: **1)** It has been established that there are Weyls points in β -WP₂ crystals and corresponding Berry curvature [Kumar, N. et al. *Nat. Commun.* 8, 1642 (2017); Zhang, K. X. et al. arXiv:2008.13553 (2020).]. In this condition, the kinetic equations of Bloch electrons naturally have the term of Berry curvature. **2)** In the previous report, it has been proposed that the interband transition leads to the generation of even-order HHG (e.g. in semiconductor GaSe). But the intensity of even-order HHG due to interband transition is much weaker than odd-order HHG [Kaneshima, K. et al. *Phys. Rev. Lett.* 120, 243903 (2018).]. And in our experiment, we can see that the intensity of even-order (for example second-order) is quite comparable to odd-order (third-harmonic generation). Therefore, interband mechanism generating the even-order plays minor role in our experiment, compared to Berry curvature mechanism. Similar discussions have been reported previously [Liu, H. Z. et al. *Nat. Phys.* 13, 262-265 (2017); Luu, T.

T. & Wörner, H. J. *Nat. Commun.* 9, 916 (2018).].

As to the power-law observed in the experiment, we double-checked fitting and the experimental data. The results indicate that the power law is not exact 2, but ranged from 2.1 to 2.5. We also found that the power-law of intensity-dependence high harmonics has been reported by some other researchers. For example, H. Liu *et al.*, *Nat. Phys.* 13, 262-265 (2016) ($\propto I_0^{3.3}$); O. Schubert *et al.*, *Nat. Photonics* 8, 119-123 (2014) (intensity of N-order HHG is deviated from $\propto I_0^N$ law). Based on the above discussions, HHG does not necessarily show a quadratic intensity-dependence and our related description in the old draft are *imprecise*. The relevant contents have been revised in the new manuscript.

In addition, both odd- and even-order harmonics are originated from the Bloch oscillations, although internal magnetic field (Berry curvature) is involved in even-order HHG. Thus, odd-order and even-order harmonic generation should follow the power-law with a similar power index. Here, we'd like to add a semi-quantitative discussion about the same power-law observed in even- and odd-order harmonic generation. The following is a brief proof.

The intensity of odd-order HHG

$$I_{i,2m+1} \propto [(2m+1)\omega_L]^2 \left| \sum_n na\varepsilon_{i,n} J_{2m+1} \left[\frac{neE_0a}{\hbar\omega_L} \right] \right|^2$$

$$\propto E_0^{2\alpha} \propto I_0^\alpha$$

In the above equation, we have used one asymptotic relationship $\frac{neE_0a}{\hbar\omega_L} > 1$,

$$J_{2m+1} \left[\frac{neE_0a}{\hbar\omega_L} \right] \propto E_0^\alpha.$$

The intensity of even-order HHG

$$\begin{aligned}
I_{2m} &\propto (2m\omega_L \frac{eE_0}{\hbar})^2 \left| \sum_n \gamma_n \left[J_{2m-1} \left(\frac{neE_0a}{\hbar\omega_L} \right) - J_{2m+1} \left(\frac{neE_0a}{\hbar\omega_L} \right) \right] \right|^2 \\
&\propto E_0^2 \left| \sum_n \gamma_n \left[J'_{2m} \left(\frac{neE_0a}{\hbar\omega_L} \right) \right] \right|^2 \\
&\propto E_0^{2+2(\alpha-1)} \propto I_0^\alpha
\end{aligned}$$

in above equation, we have used $J_{2m-1}(x) - J_{2m+1}(x) = 2J'_{2m}(x)$, and asymptotic relationship of Bessel function.

Therefore, we think that the power-index of the dependence of the intensity of odd-order HHG on the intensity of fundamental wave should be similar to that in even-order HHG. Maybe the full first-principles calculation can give a more precise power-index in this problem.

5. In Figure 4, the red lines retrieved through fitting the experimental data show jumps at the gamma points. Can you discuss this feature? Is it a fitting artifact? Add new picture.

Answer: Thanks for reviewer's comments. In Figure 4 of our old manuscript, the energy band information obtained by fitting the experimental data is mainly about the tight-binding coefficient t , and does not include the information about the top of the VB (or the bottom of the CB). The two sides of the gamma point belong to different paths, resulting in jumps at the Γ point when the parameters are optimized separately. This is *imprecise*. We have made a correction by taking the continuity at the gamma point into consideration and re-fitted the experimental results. Now the improve fitting curves were added in our new draft (see Fig. A2). Evidently, there is no jumps at the gamma points.

Fig. A2 **a** and **b**, The blue lines are the electronic dispersion curves of conduction and valence bands crossing Fermi energy ($E_F=0$) obtained by first-principle calculation. The red lines are the ones retrieved through fitting θ -dependent odd-order HHG of β -WP₂ crystals. The two results fit well. **c** and **d**, The blue lines are theoretically calculated Berry curvature of β -WP₂. The red lines are experimental Berry curvature of β -WP₂ fitted by even-order HHG, which is in good agreement with the theoretical ones.

6. In the supplement, the x-axis in Figure S3 d) is incorrectly labeled as B(T), instead of T(K).

Answer: We are very sorry for this negligence in the old manuscript. Now the Figure S3(d) is improved in our new draft (see Fig. A3).

Fig. A3 The dependence of charge carrier densities n_e and n_h , as well as carrier mobility μ_e and μ_h of electrons and holes on temperature, respectively.

Overall, I believe this is a solid and interesting study that both established unambiguously the possibility of HHG in Weyl semimetals, and also uncovers interesting puzzles for future work concerning the nature of HHG in topological materials. Assuming that the minor issues and questions raised can be addressed, I recommend this manuscript for publication.

In conclusion, we did carefully solid/elucidate our point in the new draft. We'd like to thank you again for your very constructive comments on how to refine our work, as well as how to present the work scientifically.

Reviewer #3 (Remarks to the Author):

The manuscript reports on a study of near-infrared/optical high-harmonic generation in the Weyl semimetal WP_2 . Even- and odd-order harmonic radiation up to the 10th orders is observed by exciting the material by 1900 nm laser pulses. While the observed odd harmonics are ascribed to Bloch oscillations, the even harmonics are interpreted as a

result of Berry curvatures which is characteristic for a Weyl semimetal. Nonlinear optical effects are in general a very interesting topic, thus the present study fits the scope of this journal. However, the interpretation of the experimental results has clear flaws.

Answer: Firstly, we'd like to thank you for your review time. Though the comments are quite critical but really constructive to make us re-consider/re-examine this work. As follows, we'd like to make our points more clearly and solid.

1. For many solid-state materials the observation of only odd-order harmonics (see Table S1) is guaranteed by the existence of centrosymmetry in the crystal structure. Here, WP₂ crystalizes in a non-centrosymmetric structure, thus it is natural to observe the even harmonics, even without invoking the effects of Berry curvatures.

Answer: We'd admit that we have not clearly discussed this issue carefully in the old draft. According to our literature review, there are about three mechanisms (nonlinear optical process, interband transition and Berry curvature mechanisms) for even-order harmonic generation in the materials with the non-centrosymmetric structure.

(1) In the viewpoint of nonlinear optics, if the crystals have non-centrosymmetric structure, the nonlinear electrical polarization P can be written as:

$$P = \chi^{(1)}E + \chi^{(2)}E^2 + \chi^{(3)}E^3 + \dots$$

We do can observe the high-order harmonic generation (HHG), especially even-order $\chi^{(2N)}$ is not zero if there is no inversion symmetry. And we can see that intensity of second-harmonic generation is still proportional to the square of incident intensity. But if the order is higher than two, the intensity of corresponding n-order harmonic generation will proportional to I_0^n . For example, the intensity of fourth-harmonic generation $I_4 \propto I_0^4$. In our experiment, we found that the intensity of n-order-harmonic generation in β -WP₂ crystals is approximately proportional to I_0^x , power index x ranged from 2.1 to 2.5. Obviously, it is different from 2N-power in perturbative

nonlinear optics. Therefore, we firstly rule out the possibility of the conventional nonlinear optical process in our even-order HHG experiment.

(2) In some papers, it has been proposed that the inter-band transition and successive the Bloch oscillation in conduction band leads to the generation of even-order HHG (e.g. in semiconductor GaSe). But there are three evidences violating the inter-band transition mechanism: 1) In the answer of comment 3 described at follows, we experimentally substantiated that interband transition in WP₂ is nearly negligible. Accordingly, even-order HHG coming from interband transition contributes little to experimental one. 2) the intensity of even-order HHG due to inter-band transition is much weaker than odd-order HHG [Kaneshima, K. et al. *Phys. Rev. Lett.* 120, 243903 (2018); Liu, H. Z. et al. *Nat. Phys.* 13, 262-265 (2017).]. Differently, we can see that the intensity of even-order (for example second-order) is quite comparable to odd-order (third harmonic generation) in our experiment. 3) According to symmetry, the electrical polarization along *c*-axis is finite while that of *a*-axis is zero (*a*- and *c*-axis are two in-plane directions of the surface of our β -WP₂ sample). If the even-order HHG is generated by non-inversion symmetry, in theory, even-order HHG should show a two-leaf pattern, but in our experimental, experimental even-order HHG has four-leaf shape (see Fig. 4 at main text). Therefore, even-order HHG only considering symmetry is not sufficient [Liu, H. Z. et al. *Nat. Phys.* 13, 262-265 (2017); Luu, T. T. & Wörner, H. J. *Nat. Commun.* 9, 916 (2018).].

(3) In some papers, the even harmonics generation is attributed to nonvanishing Berry curvature in the electronic bands of the materials with the non-centrosymmetric structure. In Table S1 at the supplementary information, we found 6 papers that reported the observation of even-harmonic-generation in semiconductors, 3 papers (α -quartz, MoS₂ and ZnO) have discussed the even-harmonic generation coming from the Berry curvature [Ghimire, S. et al. *Nat. Phys.* 7, 138-141 (2011); Liu, H. Z. et al. *Nat. Phys.* 13, 262-265 (2017); Luu, T. T. & Wörner, H. J. *Nat. Commun.* 9, 916 (2018).]. In our work, we took the Berry curvature mechanism for the even harmonics. There are two reasons: 1) It has been established that there are Weyls points in β -WP₂ crystals and corresponding Berry curvature by angle-resolved photoemission spectroscopy and

theoretical calculations [Kumar, N. et al. *Nat. Commun.* 8, 1642 (2017); Zhang, K. X. et al. *Chin. Phys. Lett.* **37**(9), 090301 (2020).]. In this case, the kinetic equations of Bloch electrons *naturally* have the term of Berry curvature. 2) We can see that the intensity of even-order (for example second-order) is quite comparable to odd-order (third harmonic generation) in our experiment. This feature is also used to support the even-order HHG coming from Berry curvature mechanism in some papers [Liu, H. Z. et al. *Nat. Phys.* 13, 262-265 (2017); Luu, T. T. & Wörner, H. J. *Nat. Commun.* 9, 916 (2018).].

Based on the above discussions, we think that the mechanism of even-order harmonic generation attributed to Berry curvature is the most natural explanation for our experimental observation. The relevant discussions have been added in the supplementary information of the new draft.

2. The band structure exhibits several bands crossing the Fermi surface, but only selected regions are compared to the experimental results as shown in Fig.4.

Answer: We are sorry that we have not discussed this problem carefully in the old draft. In our experiments, the infrared laser is perpendicularly incident to the surface of β -WP₂ crystals, and then the electrical field of the laser beam is on the *ac*-plane of β -WP₂ crystals, Accordingly, the *A*-vector is also along the *ac*-plane (in this work we use the radiation gauge $E = -\frac{\partial A}{\partial t}$). Therefore, Bloch oscillation can only along Γ -X, Z- Γ , and Y-X1 directions (see Fig. A4). The electronic band structure of other regions cannot be reached. These are the reasons that we only choose several regions of the Brillouin zone (see Fig. A5) to show the experimental and theoretical electronic band structure. The electronic band structures along these three directions (Γ -X, Z- Γ , and Y-X1 directions) have given in Fig. 3 at the main text and the experimentally accessible branches of electronic bands have been highlighted by red color.

Fig. A4 Electronic band structure of β -WP₂ considering spin-orbit coupling.

Fig. A5 Brillouin zone of β -WP₂. [Autès, G. et al. *Phys. Rev. Lett.* 117, 066402 (2016).]

We'd like to mention that in each either conduction- or valence-band, there are two bands of quite similar shapes, which come from splitting of two-fold degenerated bands by spin-orbit-interaction. The parameters in the tight-binding model of these two-branch conduction-/valence-bands are quite close. Numerically, two sets of tight-binding models cannot be distinguished after numerical optimization even initially we set two conduction-/valence- bands with different tight-binding-model parameters. So, we used only one band model to simulate these two-branch conduction-/valence-bands.

3. Since the energy of the laser pulses is quite high and allows interband transitions, even more bands can be involved in the dynamical processes, which however is not discussed in the manuscript.

Answer: We did not discuss this important issue at the old draft. Actually, there are several reasons that we did not discuss the contributions of interband transitions in the manuscript.

Fig. A6 Comparison of transient absorption (TA) of β -WP₂ and silicon originated from one-photon-excited interband transition.

(1) We experimentally studied the interband dynamics of β -WP₂ crystals by ultrafast transient absorption spectroscopy. Ultrafast transient absorption (TA) spectroscopy is a precise technique to investigate the interband dynamic process of materials. We performed TA measurement here to identify the probability of interband transition in β -WP₂ system using a homemade femtosecond pump-probe system. The laser source was a commercial Ti:sapphire mode-locked laser centered at 800 nm with 120-fs duration and 80-MHz repetition rate. The output laser was split into two beams, with one frequency-doubled through BBO as the pump, and the other to be the probe

beam. The sample was subjected to the two beams in the reflection geometry.

As shown in Fig. A4, along S direction, there is a main direct bandgap of ~ 0.8 eV between the valence band edge and the conduction band bottom above Fermi level. The single photon energy of 400-nm pump and 800-nm probe is sufficient to cross the band gap. In addition, this bandgap is closed to the 1.1 eV indirect bandgap of silicon, thus the transient measurement of silicon crystal was also taken for comparison. In the indirect bandgap structure of silicon, interband transition needs absorption of phonon to satisfy momentum conservation, which usually has much lower transition probability than direct bandgap. However, as shown in Fig. A6, we see the distinguished reflectivity dynamics on silicon just at pump strength of 0.3 GW/cm^2 due to the interband absorption bleaching, while no discernible response on $\beta\text{-WP}_2$ even at the maximum pump of 5.3 GW/cm^2 . It therefore demonstrates the absence of interband transition process in $\beta\text{-WP}_2$, which is possibly suppressed by the strong intraband current arising from the strong intraband absorption of the huge-concentration ($\sim 10^{21} \text{ cm}^{-3}$) free carriers originally existing on conduction band. Intraband absorption process shows few signals in TA spectroscopy, because it does not inject additional electrons into conduction band to change the total population of free electrons.

For the 1900-nm excited harmonics generation on $\beta\text{-WP}_2$ in our work, two-photon absorption mechanism is responsible for the interband transition since 1900-nm photon energy (0.65 eV) is lower than the bandgap of ~ 0.8 eV. As we know, two-photon absorption is a third-order process, whose absorption cross section is typically several orders of magnitude smaller than one-photon absorption cross section [R. W. Boyd, 2003, *Nonlinear Optics*, 2nd ed. (Academic Press, New York)]. Hence, although the maximum strength of 400-nm pump here is tens of times smaller than the 1700-nm threshold strength (0.29 TW/cm^2) for even-order HHG, the contribution of interband transition for HHG is negligible in this $\beta\text{-WP}_2$ system. Therefore, the efficient HHG in this work mainly comes from the intraband Bloch oscillation of free electron on overlapped conduction and valence bands of $\beta\text{-WP}_2$ (semimetals).

(2) Generally, in semiconductor, both the interband polarization and the intraband current could have contribution to HHG. The metals may have different situation

compared to semiconductor. Recently, Shimomura, K. et al., have reported the HHG in *metal* is mainly attributed to intraband radiation because there is a high-density of free electron in metals, quite similar to our β -WP₂ case ($\sim 10^{21}$ cm⁻³) [Shimomura, K. et al. High Harmonic Generation in Metallic Phase of 2H-NbSe₂. doi: 10.1109/IRMMW-THz.2019.8874306.]. In other words, the interband polarization would play a minor role in HHG of metals. As shown in Fig. A7, the electronic band structure of β -WP₂ is sure a typical *semimetal* within a large energy range (-7.0~7.0 eV).

Fig. A7 Electronic band structure of β -WP₂ considering spin-orbit coupling with the energy range of -7.0~7.0 eV.

(3) Theoretically, the majority of the population during the laser pulse is concentrated in the first (lowest) conduction band and falls nearly exponentially with the band number [Luu, T. T. et al. *Nature* 521, 498-502 (2015).]. Therefore, the Bloch oscillations of those higher conduction bands above Fermi level to HHG can be neglected.

Based on the above-discussions, we believe that inter-band transition is not the major factor to explain our experimental observation. Now, the relevant discussions are added to the supplementary information at the new draft. Certainly, we really wish theoretical colleagues to do the calculation based on time-dependent first-principles calculation to elucidate HHG in Weyl semimetal β -WP₂ crystals.

With these considerations I cannot recommend the present manuscript for publication in Nature Communications.

In summary, we'd like to thank you for your constructive comments on our draft. We wish our explanations can elucidate your comments. Really thank you.

Reviewers' comments:

Reviewer #1 (Remarks to the Author):

The authors have addressed my concerns. I would now recommend the publication of this manuscript in Nature Communications.

Reviewer #3 (Remarks to the Author):

The authors have indeed spent efforts to answer my questions in their response letter, and also made modifications in the revised manuscript and SI. However, my major concerns have not been convincingly addressed.

1. Symmetry consideration is independent on the perturbative or nonperturbative nature of the dynamic processes. Indeed the experiment revealed a nonperturbative regime for the observed HHG, but solely based this one cannot claim that the even-order harmonics are not primarily due to the breaking of inversion symmetry. On the phenomenological level, for example, there is no a priori reason to omit the sine-function terms in the equation in Line 165. It is neither obvious why the cosine terms should be absent in Line 184.

2. It is unclear how the 400nm pump 800nm probe measurement can rule out an interband transition excited by the 1900nm pulse. For example, the 1900nm photon is sufficient to excite an interband transition close to the Z point along the T-Z direction. Along the γ -Y direction the energy is also sufficient to excite an electron to higher conduction bands.

3. In the response various references were mentioned to support the claims of the authors, but most of them are not necessarily relevant to the specific Weyl system treated here. For example, the authors argued that " the intensity of even-order (for

example second-order) is quite comparable to odd-order (third harmonic generation) in our experiment. This feature is also used to support the even-order HHG coming from Berry curvature mechanism in some papers [Liu, H. Z. et al. Nat. Phys. 13, 262-265 (2017); Luu, T. T. & Wörner, H. J. Nat. Commun. 9, 916 (2018).]". However, Liu et al dealt with a monolayer sample which has centrosymmetry in bulk, which is very different to the situation here. Moreover, isn't it more natural to ascribe the comparable intensity of the even- and odd-harmonics to the same mechanism?

To summarize, the flaws of the interpretation remains. It is not convincing to assign the observed even- and odd-harmonics to completely different mechanisms. Without addressing the raised points convincingly in the manuscript, I cannot recommend publication in Nature Communications.

Response Letter to the Reviewers

Reviewer #1 (Remarks to the Author):

The authors have addressed my concerns. I would now recommend the publication of this manuscript in Nature Communications.

Answer: We thank the reviewer for accepting our manuscript.

Reviewer #3 (Remarks to the Author):

The authors have indeed spent efforts to answer my questions in their response letter, and also made modifications in the revised manuscript and SI. However, my major concerns have not been convincingly addressed.

Answer: we'd like to thank you for your review time. These comments are critical and quite useful for us to re-consider our conclusion more deeply and present our data more clearly. Accordingly, we did further experimental work and clarifications to address these comments. All revises in the main text have been highlighted by red color. The replies to these questions are outlines as follows.

Before the detailed answer the comments, we'd like to point out the two points that are not clearly in the old draft and last reply.

(1) We have not clearly presented the *crucial* role of breaking of inversion-symmetry in even-order HHG. Actually, breaking of inversion symmetry gives rise to the Weyl semimetal state in β -WP₂ crystals. Accordingly, there is “spike-like” Berry curvature in β -WP₂ crystals. In turn, there is significant even-order HHG here.

(2) Both odd- and even-order HHG in β -WP₂ come from the Bloch oscillation. Under laser acceleration, Bloch electrons are accelerated, this motion generates the normal

odd-order HHG; simultaneously, if there is finite Berry curvature, under effect of Berry curvature (equivalent to an internally spontaneous magnetic field), electrons will have an additional anomalous velocity term (cyclotron movement). This anomalous velocity term leads to even-order HHG, **which still belongs to Bloch oscillation**. The schematic of above two-process is shown in Fig. 1.

Fig. 1. The schematic showing the electron's Bloch oscillation under both external electric field E of laser beam and Berry curvature $\Omega_y(\mathbf{k})$. v_{\parallel} represents the velocity parallel to electric field E , while v_{\perp} does the anomalous velocity term coming from Berry curvature $\Omega_y(\mathbf{k})$. v_{\parallel} and v_{\perp} give rise to odd-order and even-order HHG, respectively. Corresponding mathematic formulas can be found in equation (1)-(4) at the main text and section 10 of supplementary information.

1. Symmetry consideration is independent on the perturbative or nonperturbative nature of the dynamic processes. Indeed the experiment revealed a nonperturbative regime for the observed HHG, but solely based this one cannot claim that the even-order harmonics are not primarily due to the breaking of inversion symmetry. On the phenomenological level, for example, there is no a priori reason to omit the sine-function terms in the

equation in Line 165. It is neither obvious why the cosine terms should be absent in Line 184.

Answer: We do agree with referee that symmetry consideration is independent on the perturbative or non-perturbative nature of dynamic process. And we find that our presentation about the role of breaking-inversion symmetry does have problem at the old draft and last reply to the reviewers' comments.

Actually, breaking inversion-symmetry has close relationship to significant Berry curvature in β -WP₂. Here breaking inversion-symmetry gives rise to the Weyl semimetal state in β -WP₂, and accordingly there is a "spike-like" Berry curvature in β -WP₂ crystals. In other words, here even-order HHG can be attributed to "spike-like" Berry curvature in Weyl semimetal β -WP₂ that is resulted from breaking of inversion-symmetry. Therefore, we emphasized the role of breaking of inversion symmetry in the revised draft. The revised parts have been highlighted by red color in the main text.

As to the second question, we have made the triangular Fourier series expansions of the electronic band structure along Γ -X- Γ and Γ -Z- Γ calculated by the first-principles local-density-approximation (LDA) method. The expansions are shown in Fig. 2. One can see obviously that the we can nearly completely ignore the sine-function term in equation of Line 165. Similarly, we also made the triangular Fourier series expansions of the Berry curvature along Γ -X- Γ and Γ -Z- Γ calculated by LDA and Wannier function methods, and the results are presented at Fig. 3. One can see again that we can ignore the cos-function term in Berry curvature.

Actually, under the protection of the time-reversal symmetry in β -WP₂ (it is a non-magnetic compound), eigenvalue of energy $E(k)$ is the *even-function* of Bloch-wavevector k , as well as Berry curvature $\Omega(k)$ is the *odd-function* of Bloch-wavevector k [A. Bohm, A. Mostafazadeh, H. Koizumi, Q. Niu and J. Zwanziger, *The geometric phase in quantum systems*, (Springer-Verlag Berlin Heidelberg 2003), page 285].

Therefore, equations in Line 165 and Line 184 in the original manuscript are accurate.

Fig. 2. Using both sine and cosine functions to simulated the electronic band structures calculated by LDA method. **One can see that the coefficients before sine functions terms are too, compared with those of cosine terms.**

a, The triangular Fourier series expansion of the valence band along Γ -X- Γ with

expansion:
$$E_{V1} = -0.4864 + 0.3310 \cos(ka) - 0.5446 \cos(2ka) - 0.2062 \cos(3ka) + \dots$$

$$+ 7.7 \times 10^{-18} \sin(ka) + 2.0 \times 10^{-18} \sin(2ka) - 1.8 \times 10^{-17} \sin(3ka) + \dots$$

b, The triangular Fourier series expansion of the conduction band along Γ -X- Γ with

expansion:
$$E_{C1} = 0.7774 + 0.3661 \cos(ka) - 0.7713 \cos(2ka) - 0.0057 \cos(3ka) + \dots$$

$$- 2.7 \times 10^{-17} \sin(ka) - 3.9 \times 10^{-17} \sin(2ka) - 4.0 \times 10^{-17} \sin(3ka) + \dots$$

c, The triangular Fourier series expansion of the valence band along Γ -Z- Γ with

expansion:
$$E_{V2} = -0.4367 - 0.4046 \cos(ka) - 0.0958 \cos(2ka) + 0.0206 \cos(3ka) + \dots$$

$$+ 3.2 \times 10^{-17} \sin(ka) + 4.5 \times 10^{-17} \sin(2ka) + 3.2 \times 10^{-17} \sin(3ka) + \dots$$

d, The triangular Fourier series expansion of the conduction band along Γ -Z- Γ with

expansion:
$$E_{C2} = 0.5345 - 0.3146 \cos(ka) + 0.0235 \cos(2ka) - 0.0147 \cos(3ka) + \dots$$

$$- 3.3 \times 10^{-18} \sin(ka) - 5.6 \times 10^{-18} \sin(2ka) - 6.8 \times 10^{-18} \sin(3ka) + \dots$$

Fig. 3. Using both sine and cosine functions to simulated the Berry curvature of β -WP₂ calculated by LDA method. **One can see that the coefficients before cosine functions terms are too, compared with those of sine terms.**

a, The triangular Fourier series expansion of the Berry curvature along Γ -X- Γ with

expansion:
$$\Omega_y(k) = 0.0000 + 6.5434 \sin(ka) - 7.2674 \sin(2ka) + 8.6544 \sin(3ka) + \dots$$

$$+ 4.9 \times 10^{-7} \cos(ka) + 3.2 \times 10^{-7} \cos(2ka) + 3.6 \times 10^{-7} \cos(3ka) + \dots$$

b, The triangular Fourier series expansion of the Berry curvature along Γ -Z- Γ with

expansion:
$$\Omega_y(k) = 0.0000 + 10.8340 \sin(ka) - 9.0915 \sin(2ka) + 2.3641 \sin(3ka) + \dots$$

$$- 1.7 \times 10^{-7} \cos(ka) - 1.3 \times 10^{-7} \cos(2ka) + 2.2 \times 10^{-7} \cos(3ka) + \dots$$

2. It is unclear how the 400nm pump 800nm probe measurement can rule out an interband transition excited by the 1900nm pulse. For example, the 1900nm photon is sufficient to excite an interband transition close to the Z point along the T-Z direction. Along the γ -Y direction the energy is also sufficient to excite an electron to higher conduction bands.

Answer: We do admit that it is too hasty for us to rule out the interband transition excited by the 1900 nm pulse. And we do agree with you that there is possible electron transitions close Z point along the T-Z direction and along γ -Y direction, as seen from electronic band structure.

To make the work more rigorous, we directly measured the dynamic process of interband transient absorption (TA) **probed at 1900 nm** (~100 fs pulse duration, 1 kHz

repetition rate) with pumped at 800 nm (800-nm photons can excite all transition whose energy is lower than 1.55 eV) in a femtosecond pump-probe system. If there is obvious interband transition at the points along the T-Z direction and along gamma-Y direction under 1900 nm photons, clear decay in TA response on 1900-nm probe pulse should be detected. However, as shown in Fig. 4, we did not detect any discernible decay response at 1900 nm, even under the pump strength of 6.2 GW/cm^2 . For comparison, we also carried out the TA at several shorter probe wavelengths under the same 800-nm pump intensity, as shown in Fig. 4. One can see the distinct absorption response at 1000 nm, which means that there is stronger absorption than 1900 nm case. **These results confirm the weak interband transition at 1900 nm.**

To cross-check above experimental data, we calculated the imaging part $\varepsilon_1(E)$ of dielectric constant $\varepsilon(E)$ (E is the energy, and can be converted to wavelength of electromagnetic-wave by timing 1240 nm/eV) by first-principles LDA method. The data is presented in Fig. 5. One can see that the ε_1 at 0.65 eV (1900 nm) is as small as 5.2, in contrast, there are a giant Drude peak at zero-energy and ε_1 being as large as 20.0 at 2.1 eV. We also can see that theoretical ε_1 is in line with TA experiment shown in Fig. 4, absorption at 1000 nm is larger than that at 1900 nm.

Combining above two data, we do believe that the inter-band transition of $\beta\text{-WP}_2$ is quite weak and its role in observed HHG is immaterial.

Fig. 4. Comparison of transient absorption of β -WP₂ at different wavelengths. One can see that the small Δ absorption at 1900 nm. The 800-nm pump intensity is 6.2 GW/cm², and the probe intensities at different wavelengths are fixed at 0.6 GW/cm².

Fig. 5. Theoretical imaginary part of dielectric constant ϵ_{imag} -energy (E) relationship. One can see that at this energy range, dielectric absorptions have two peaks, one is the Drude peak at 0-energy ($\epsilon_{imag}=30$); the other is the peak at 2.1 eV ($\epsilon_{imag}=20.0$). ϵ_{imag} at 0.65 eV (corresponding electromagnetic-wavelength of 1900 nm used in our HHG experiment) is around 5.2.

3. In the response various references were mentioned to support the claims of the authors, but most of them are not necessarily relevant to the specific Weyl system treated here. For example, the authors argued that "the intensity of even-order (for example second-order) is quite comparable to odd-order (third harmonic generation) in our experiment. This feature is also used to support the even-order HHG coming from Berry curvature mechanism in some papers [Liu, H. Z. et al. Nat. Phys. 13, 262-265 (2017); Luu, T. T. & Wörner, H. J. Nat. Commun. 9, 916 (2018).]". However, Liu et al dealt with a monolayer sample which has centrosymmetry in bulk, which is very different to the situation here. Moreover, isn't it more natural to ascribe the comparable intensity of the even- and odd-harmonics to the same mechanism?

Answer: We admit that we have not discussed this problem clearly in the last reply, which may be due to our poor language ability.

Actually, the aim in citing these papers is to express that **authors in these works used the Berry curvature to explain the observed the HHG**. The similarity between our work and cited works is that **there is finite Berry curvature in these systems, rather than samples used in cited paper are Weyl semimetals**. The only difference in our work is that the Berry curvature shows “spike-like” feature in our case (β -WP₂) that is a Weyl semimetal, but Berry curvature is a much smooth function in the samples in cited papers. The comparison of Berry curvature in our and cited works is shown in Fig. 6. Evidently, the Berry curvature in cited works is smoother/smaller than that in our case. Citing “Liu et al work of monolayer MoS₂” is also because in *mono-layer* MoS₂, there are broken-spatial-inversion-symmetry and finite Berry curvature, these features are quite similar to our β -WP₂. We do not want to compare our β -WP₂ to *bulk* MoS₂.

One more thing we’d like to mention is that: in the samples in the cited papers, **the crystal structures of these samples all have broken-spatial-inversion-symmetry**. This feature is the same as our sample (β -WP₂).

As to “Moreover, isn't it more natural to ascribe the comparable intensity of the even- and odd-harmonics to the same mechanism?” comment, based on equations (1)-(4) at main text of this manuscript, even-order harmonic generation can be explained as Bloch oscillation **under “spike-like” Berry curvature**, while odd-one is attributed to Bloch oscillation only considering the acceleration of electron along electronic bands under stimulus of laser beam (see schematic of Fig. 1 in this reply). These two mechanisms are described within the *same quasi-classical dynamics* [A. Bohm, A. Mostafazadeh, H. Koizumi, Q. Niu and J. Zwanziger, *The geometric phase in quantum systems*, (Springer-Verlag Berlin Heidelberg 2003), chapter 12.3].

Fig. 6. **a**, Berry curvature of the conduction band for monolayer MoS₂ [Liu, H. Z. et al. Nat. Phys. 13, 262-265 (2017)]. **b**, The fitted Berry curvature of α -quartz [Luu, T. T. & Wörner, H. J. Nat. Commun. 9, 916 (2018)]. **c** and **d**, The blue lines are the theoretically calculated Berry curvature of β -WP₂. The red lines are the experimental Berry curvature of β -WP₂ fitted by even-order HHG.

To summarize, the flaws of the interpretation remains. It is not convincing to assign the observed even- and odd-harmonics to completely different mechanisms. Without addressing the raised points convincingly in the manuscript, I cannot recommend publication in Nature Communications.

Answer: In summary, in the new draft we do have emphasize the role of broken inversion-symmetry in even-order HHG. It is the broken inversion-symmetry generating Weyl state in β -WP₂ crystal; and even-order harmonic generation can be explained as Bloch oscillation under “spike-like” Berry curvature, while odd-one is attributed to Bloch oscillation. **The even- and odd-order HHG in β -WP₂ can be explained within the same quasi-classical dynamics of Bloch electrons.**

We wish our new experiments and clarifications address your comments.

REVIEWER COMMENTS

Reviewer #3 (Remarks to the Author):

If one plots the dispersion relation along a specific a-b-a direction of high symmetry (such as Γ -z- Γ in Fig.2 of the "Response to Referees Letter"), one should be surprised not to get an even function. In reality the band structure is of three dimension. Is anywhere parallel this high-symmetry direction a perfect even function? I do not insist on further revisions, but leave this to the authors' decision. I am fine if this work is published together with the response letter.

Response Letter to the Reviewers

Reviewer #3 (Remarks to the Author):

If one plots the dispersion relation along a specific a-b-a direction of high symmetry (such as Γ -z- Γ in Fig.2 of the "Response to Referees Letter"), one should be surprised not to get an even function. In reality the band structure is of three dimension. Is anywhere parallel this high-symmetry direction a perfect even function? I do not insist on further revisions, but leave this to the authors' decision. I am fine if this work is published together with the response letter.

Answer: Thanks for reviewer's comments. We are very sorry for the un-clear description for this problem at the last Response letter. This problem comes from our un-clear description/definition of high-symmetry points. Actually, we crossed the **two adjacent Brillouin zones** to choose k and $-k$ (k is the Bloch wavevector) points to check whether electronic band structures $E(k)$ are even-function of k . Here we have corrected the figure 5 of the main text (see Fig. 5 in the revised manuscript) and the figure 2 of the last "Response to Referees Letter" (see Fig. 1 here). Point Z, as well as X and X1, is the boundary of the Brillouin zone. Point Γ represents the center of the Brillouin zone, and point Γ' is the center of **another adjacent Brillouin zone**. As labeled in Figs. 1c-d and Fig. 2, Γ -Z and Z- Γ' are two different paths with center inversion symmetry in the k -space. One can see clearly that eigenvalue energy $E(k)=E(-k)$ (k is the Bloch wavevector) along the high-symmetry lines, therefore we can expand the $E(k)$ as Fourier series only consisting of cosine terms.

As to the question "Is anywhere parallel this high-symmetry direction a perfect even function?", generally it is not. If chosen k -line does not cross the Γ -point ($k=0$), though it is parallel to the high-symmetry direction, we cannot find k and $-k$ points in this specific line. Therefore, we cannot determine whether it is a perfect even function of k . But if chosen k -line cross the Γ -point ($k=0$), $E(k)$ is definitely an even function of

k , which is guaranteed by time-reversal-symmetry in β -WP₂ and numerically confirmed by the first-principles calculations.

And more, we do agree to publish our manuscript together with the whole response letter of the reviewer.

Fig. 1. Using both sine and cosine functions to simulated the electronic band structures calculated by LDA method. **One can see that the coefficients before sine function terms are too small, compared with those of cosine terms.** The definitions of X, X1, Γ and Γ' can be found in the Fig. 2. Obviously, we can see that eigenvalue energy $E(k)=E(-k)$ (k is the Bloch wavevector) along the high-symmetry lines.

a, The triangular Fourier series expansion of the valence band along Γ -X-X1- Γ' with

$$\begin{aligned} \text{expansion: } E_{V1} = & -0.4864 + 0.3310 \cos(ka) - 0.5446 \cos(2ka) - 0.2062 \cos(3ka) + \dots \\ & + 7.7 \times 10^{-18} \sin(ka) + 2.0 \times 10^{-18} \sin(2ka) - 1.8 \times 10^{-17} \sin(3ka) + \dots \end{aligned}$$

b, The triangular Fourier series expansion of the conduction band along Γ -X-X1- Γ' with

$$\begin{aligned} \text{expansion: } E_{C1} = & 0.7774 + 0.3661 \cos(ka) - 0.7713 \cos(2ka) - 0.0057 \cos(3ka) + \dots \\ & - 2.7 \times 10^{-17} \sin(ka) - 3.9 \times 10^{-17} \sin(2ka) - 4.0 \times 10^{-17} \sin(3ka) + \dots \end{aligned}$$

c, The triangular Fourier series expansion of the valence band along Γ -Z- Γ' with

expansion:
$$E_{V_2} = -0.4367 - 0.4046 \cos(ka) - 0.0958 \cos(2ka) + 0.0206 \cos(3ka) + \dots$$

$$+ 3.2 \times 10^{-17} \sin(ka) + 4.5 \times 10^{-17} \sin(2ka) + 3.2 \times 10^{-17} \sin(3ka) + \dots$$

d, The triangular Fourier series expansion of the conduction band along Γ -Z- Γ' with

expansion:
$$E_{C_2} = 0.5345 - 0.3146 \cos(ka) + 0.0235 \cos(2ka) - 0.0147 \cos(3ka) + \dots$$

$$- 3.3 \times 10^{-18} \sin(ka) - 5.6 \times 10^{-18} \sin(2ka) - 6.8 \times 10^{-18} \sin(3ka) + \dots$$

Fig. 2. **a**, Brillouin zone of β -WP₂. **b**, Two Brillouin zones of β -WP₂. The Γ -Z and Z- Γ' are the paths as shown in Figs. 1c-d. We extended the electronic band structure calculation extending to neighboring Brillouin zone (BZ) to check whether $E(k)$ is the even function of Bloch wavevector k .

Wish the above-description answer this question!

Really thank you for your comments on how to solid and refine our draft.